# StarHopper: A Touch Interface for Remote Object-Centric Drone Navigation

Jiannan Li*        Ravin Balakrishnan†        Tovi Grossman‡

University of Toronto

## ABSTRACT

Camera drones, a rapidly emerging technology, offer people the ability to remotely inspect an environment with a high degree of mobility and agility. However, manual remote piloting of a drone is prone to errors. In contrast, autopilot systems can require a significant degree of environmental knowledge and are not necessarily designed to support flexible visual inspections. Inspired by camera manipulation techniques in interactive graphics, we designed StarHopper, a novel touch screen interface for efficient object-centric camera drone navigation, in which a user directly specifies the navigation of a drone camera relative to a specified object of interest. The system relies on minimal environmental information and combines both manual and automated control mechanisms to give users the freedom to remotely explore an environment with efficiency and accuracy. A lab study shows that StarHopper offers an efficiency gain of 35.4% over manual piloting, complimented by an overall user preference towards our object-centric navigation system.

**Index Terms:** Human-centered computing—Human Compute Interaction (HCI)—Interaction Techniques;

## 1 INTRODUCTION

Researchers in telepresence have long envisioned 'beyond being there' [23]. Replicating all relevant local experiences, while remote, should not be the only goal of telepresence; rather, we should also strive to create telepresence systems which can enable benefits that are not possible when the person is physically present. As such, telepresence goes from replication to augmentation. One particular instance of this vision is enabled by camera drones: our local bodies can only walk on the ground, but our remote bodies can fly.

Current commercial remote robotic presence platforms have mostly been designed to replicate face-to-face conversation experiences [42, 59, 60]. Researchers exploring their usage in various scenarios have noted a number of social and functional issues due to their insufficient mobility [2, 21, 59]. Unmanned micro aerial vehicles (called 'drones' hereafter) have already been applied in industrial inspection settings, such as aircraft surface checks [3]. With drones becoming more affordable and reliable, they hold the potential for enabling more flexible remote presence and visual inspection experiences [18, 22, 27, 43, 52, 60] for the general population.

While drones offer promise for such telepresence applications, they are challenging to manually control remotely, due to numerous factors including high degrees of freedom, narrow camera field-of-views, and network delays [44]. Their control interfaces - virtual or physical joysticks for consumer drones - are also unfamiliar for many users and take extended training time to master [52].

To relieve the burden of manual piloting, autopilot techniques have been applied to drone control. Most existing drone autopilot

---

*e-mail: jiannanli@dgp.toronto.edu

†e-mail:ravin@dgp.toronto.edu

‡e-mail:tovi@dgp.toronto.edu

Graphics Interface Conference 2020
28-29 May

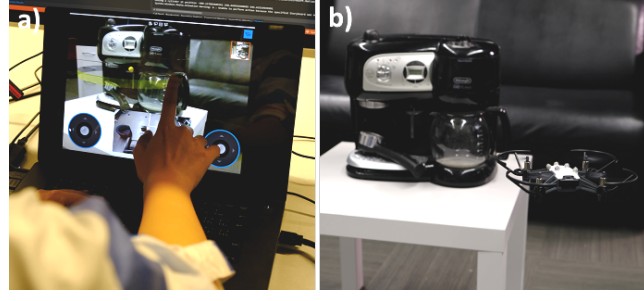

Figure 1: Operating a camera drone remotely to inspect an apartment. (a) The user specifies a desired view of the coffee machine by dragging on the drone's camera view (b) the drone flies towards to the specified viewpoint.

interfaces are based on specifying a series of planned waypoints in a 2D or 3D global map [12, 15, 40, 52]. However, in a situation where a user wishes to perform a real-time inspection, setting waypoints a priori may not be efficient for producing the viewer's desired viewpoints. For example, the viewer may wish to see something from a closer distance, from a different viewpoint, or view an area they didn't know about when the waypoints were set. Some autonomous systems avoid the use of waypoints and execute higher-level plans [13, 28, 36] , such as following a subject to form canonical shots [28], but they typically do not offer the flexibility for exploring remote environments.

The difficulty of drone piloting poses a significant barrier for the widespread adoption of free-flying robots. The goal of this research is to design a camera drone control interface to support efficient and flexible remote visual inspection for now universally adopted touchscreens. Inspired by recent work in semi-autonomous hybrid systems [36, 44], we wish to combine the strengths of both manual and automatic piloting into a single hybrid navigation interface. Our work is also inspired by decades of research in interactive graphics, for which many camera navigation techniques have been established [6, 20, 31, 37]. Most relevant, we build upon object-centric techniques, where zooming, panning, and orbiting occurs relative to the location of a 3D object of interest.

Existing object-aware drone navigation interfaces, such as DJI ActiveTrack [9] and XPose [36], have been designed for aerial photography within visual line-of-sight. As such, they lack support for two important requirements of remote navigation and inspection: first, free exploration of a remote environment, which may include objects out of the initial camera field-of-view; and second, flexible inspection from various viewing angles or distances in relation to the object-of-interest.

We propose StarHopper, a remote object-centric camera drone navigation interface that is operated through familiar touch interactions and relies on minimal geometric information of the environment (Fig. 1). The system is designed based on a set of design goals for remote object-centric drone navigation. It consists of an overhead camera view for context and a 3D-tracked drone's first-person view

for focus. New objects of interest can be specified through simple touch gestures on both camera views. We combine automatic and manual control via four navigation mechanisms that can complement each other with unique strengths, to support efficient and flexible visual inspection. The system focuses on indoor environments, representative of tasks such as remote warehouse inspection [52] and museum visits [54], and where positional tracking technology is more reliable.

A remote object inspection study showed that StarHopper was 35.4% faster than a manual baseline interface, for both simple and complex navigation routes. Users expressed general preferences towards object-centric navigation for remote inspection. We conclude by discussing potential design opportunities and future research to further increase the efficiency of remote visual inspection tasks.

## 2 RELATED WORK

Our work builds upon prior research in interactive drones, robotic teleportation and camera navigation techniques in computer graphics.

### 2.1 Interactive Drones

Recently, interacting with drones has received considerable attention in the HCI community. Researchers have explored their various roles they can take on, such as programmable matter [18], haptic devices [1, 57], mobile navigation guides [5, 32], and outdoor exercise companions [19]. The high mobility and agility of drones naturally lend themselves to serving as flying cameras, and many mid-to-high range consumer drones were designed for video capture.

Given the difficulty of manually piloting a drone, there has been prior research aimed to partially or fully automate flight paths to reduce piloting efforts [13, 40] or to improve camera shot quality via computationally optimized flight paths [15, 16]. Some work has also considered real-time adjustment to autopilot plans to adapt to changes [52]. The majority of the autopilot interfaces require the user to set flight waypoints on an exocentric 2D or 3D map.

Other work has aimed to further reduce user input through acting more autonomously based on environmental knowledge [13, 24, 28, 36, 39]. Notably, several aerial photography interfaces leverage the positional and geometrical information of their photography subjects. DJI drones with ActiveTrack [9] can follow a subject, either automatically recognized or specified through a touchscreen gesture, and orbit around the subject in response to user joystick input. Huang et al. presented an interface by which the user could move the drone relative to the recognized skeleton of a person [24]. With XPose [36], the user starts from choosing one of the candidate positions for the drone to photograph a selected subject and then fine-tune the framing through touch gestures on the camera view. However, all of them assume a single subject that is already in the camera field-of-view, and trade navigation flexibility for fast and visually pleasing composition.

Another intuitive approach for remote drone control is to map the locomotion and head orientation of the operator to the locomotion and pose of the drone [22, 60]. However, with the current state of technology, the delay between human and drone movement is significant and detrimental to such experiences. Our design draws on prior research on camera drone navigation, combining the efficiency of auto-piloting with the flexibility of semi-manual and manual controls. Notably, our system does not require a 3D environment map and supports object-centric navigation, were users can position the drone relative to the object of interest, rather than directly in the world space.

### 2.2 Object-Aware Robot Operation

Many interfaces have been developed for instructing robots to interact with objects in the environment. Human-robot interaction researchers have created interfaces that enable users to direct a robot

to a target object and specify the kinds of actions that should be performed. Kemp et al. [29] utilized a laser pointer for designating objects of interest for a mobile robot, which would turn to the pointed object. Ishii et al. [25] further explored mapping three actions of a house robot to a set of laser pointer gestures around the objects. Kent et al. [30] studied the performance of a teleoperation interface that allowed users to directly specify a robotic arm grasping pose on the 3D scan of an object.

Prior object-aware robot interfaces were mostly designed for simple physical manipulation, like grasping or moving objects, which typically have well defined ideal end results. In contrast, visual inspection can involve more free exploration and unplanned movement. Our system extends existing object-aware robot control to camera drones, through a set of mechanisms to work under such ambiguity.

### 2.3 Camera Navigation in Computer Graphics

Common 3D navigation tools offered by commercial modelling software include pan, zoom, and orbit [31]. Mastering these tools can require extensive learning and even experienced computer users can find 3D navigation confusing [11].

As such, numerous techniques have been developed to improve camera navigation [8]. In particular, object-centric techniques improve navigation by taking into account the user's object of interest. For example, HoverCam allowed users to move a virtual camera smoothly around the edges of an object of interest [31]. StyleCam supported designing camera trajectories with certain stylistic elements around an 3D object [6]. Navidget enabled the user to specify a viewing angle around a 3D point of interest using 2D input [20]. Other techniques include clicking to approach and focus on an object [20, 37] and through-the-lens camera control [17], where a user manipulates a virtual camera by controlling features seen in the image through its lens.

We took inspiration from classical object-centric camera control in computer graphics and designed a set of camera manipulation techniques for physical inspection. We adapt these classic techniques to account for inaccurate measurements of the real world, and the inherent limitations of aerial robotic systems.

## 3 DESIGN GUIDELINES

Grounded by our review of prior literature, we now present a set of guidelines for remote object-centric drone navigation.

### 3.1 Support Situation Awareness

A user's knowledge about the state of the surrounding environment, known as situation awareness, is a key factor for successful teleoperation [58]. Users of current drone autopilot systems rely on 2D or 3D maps of the remote space to maintain basic environmental awareness for flight planning [40, 44, 52]. Such maps are mostly static and unable to reflect the changes in a dynamic environment. Up-to-date situational awareness can be more relevant to a user for remote drone inspection, as the high mobility of drones can be better exploited in environments with changing situations.

### 3.2 Minimize Reliance on Environmental Information

While 3D reconstruction technology has been advancing rapidly (e.g. [26, 61] ), it is still a computationally expensive process and its performance is subject to environmental conditions. As such, we argue for minimizing the reliance of our interface design on environmental knowledge. A navigation interface that minimizes its assumptions about available environmental information should be able to adapt to a wider range of real-world applications.

### 3.3 Combine Automated and Manual Control

The navigation system should support both efficient and flexible viewpoint control. Some automated control mechanisms can infer

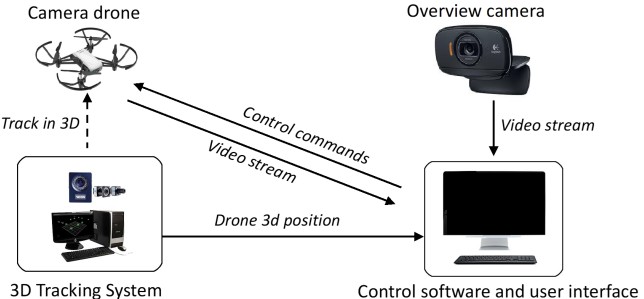

Figure 2: StarHopper system components.

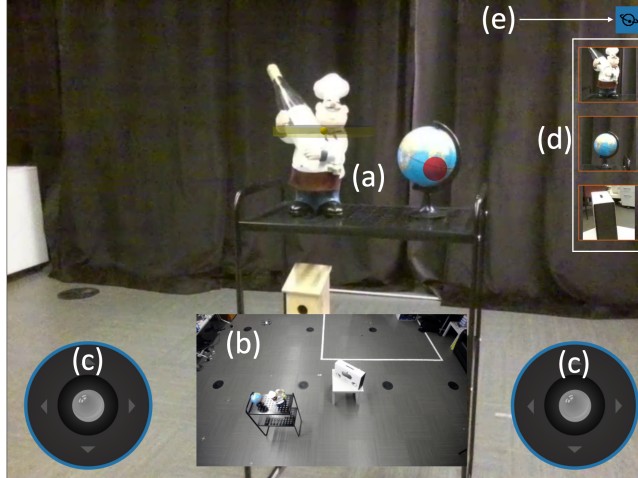

Figure 3: The StarHopper user interface. (a) Remote drone camera view. (b) Overview camera view. (c) Virtual joysticks. (d) Object-of-interest list. (e) Icon for object-centric mode.

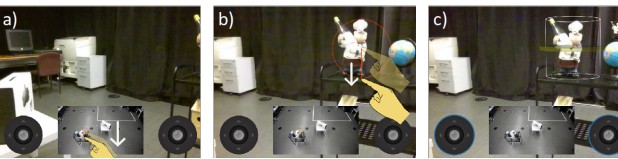

Figure 4: The object-of-interest registration procedure. (a) The user selects the object of interest in the overview camera view with a drag gesture. (b) The drone points to the region, and the user then selects the object again in the drone camera view.

a user's intent to simplify interaction. But this efficiency gain may come at the price of control flexibility, which is important for users to make adjustments when the automated control fails to achieve the desired view, or the user simply wishes to make real-time adjustments to a navigation path. Recent research has shown that users prefer having some level of manual control even when the automation accuracy is high [48]. Control flexibility also supports free exploration, an inherent component in many remote inspection scenarios, such as exploring a museum [55]. Thus, it is desirable to investigate the design of semi-autonomous, or hybrid interfaces, so that users can gain the efficiency benefits from automation and at the same time retain control flexibility.

### 3.4 Support Simple Touch Interactions

As many drone control applications are now available on mobile devices, we focus on touch-based interactions. Within this context, navigation commands should utilize known interaction patterns and simple touch gestures to improve efficiency and facilitate learning. Even though simple gestures may not have the capacity to encode complex navigation paths, a complimentary automated control system should be able to make reasonable assumptions about the user's intent, based on the input and its knowledge about the object of interest.

### 3.5 Respect Physical Constraints

In comparison with a virtual camera, the control of a physical drone may suffer from technical limitations. For example, navigation transitions cannot be immediate, latency may be present in the control system, and there may be drift, inaccuracy, or instability in the navigation paths and viewpoints. Furthermore, unlike virtual camera control, a physical control needs to avoid objects along its path. When designing interactions for camera drones, these physical constraints should be accounted for.

## 4 STARHOPPER

### 4.1 System Overview

Based on the design guidelines described above, we built StarHopper, a remote drone navigation interface that implements an object-centric control paradigm. We now describe its user interface and relevant hardware configuration (Fig. 2).

Prior research has shown the effectiveness of an live exocentric overview for enhancing situation awareness in robot teleoperation [10, 49, 53]. To help the user maintain situation awareness, we installed a regular RGB camera at a vantage point to provide an overview of the remote environment, in a similar fashion to [50, 54]. The position and orientation of the drone in the remote space is tracked in real-time to enable autopiloting and object-centric controls.

StarHopper provides a touch screen interface for the users to view the drone's live stream video and to perform drone navigations

(Fig. 3). The drone camera feed fills the screen. The overview camera video and two virtual joysticks are at the bottom of the interface.

### 4.2 Registering an Object of Interest

With StarHopper, the user can obtain the approximate position and dimensions of an object through a simple two-step procedure, without using pre-built maps or expensive real-time 3D reconstruction methods. The user first selects the object of interest in the overview camera view through a drag gesture (Fig. 4a). The starting point of the gesture defines the center of a circle and the drag distance defines the radius. When the gesture is completed, the drone turns to look at the selected region. The user then performs another drag gesture, this time in the drone camera view, to select the same object (Fig. 4b). A computer vision algorithm triangulates the position of the object from these two regions and estimates the dimensions of a bounding cylinder of the object. The technical details of this procedure are described later.

### 4.3 Navigation Mechanisms

Once an object is registered, it is immediately set as the object-of-interest. The center of the object-of-interest is marked in the drone camera view using a small gold 3D sphere. Inspired by camera control mechanisms in interactive graphics, we have designed three object-centric physical camera navigation mechanisms for viewing an object of focus: *360 viewpoint widget*, *delayed through-the-lens control*, and *object-centric joysticks*.

### 4.4 360 Viewpoint Widget

The *360 viewpoint widget* is a widget for quickly navigating to and focusing on an object of interest, from a user-specified viewing angle. The widget takes the shape of a semi-transparent 3D ring, surrounding the bounding cylinder of the focus object (Fig. 5a). A user taps the ring to activate the widget. A 3D arrow aimed at the ring appears next to the user's touch point, indicating the desired viewing direction.

While the finger is down, the user can drag horizontally to rotate the arrow around the ring, to specify the desired viewing angle. The user can also drag the finger vertically to move the ring up or down to adjust the vertical height of the viewpoint (Fig. 5b). Once the user releases the finger, the autopilot system moves the drone to the calculated viewpoint.

The algorithm determines a reasonable default viewing distance, based on the size of the bounding cylinder. The distance is set so that the object can fit into the central 1/3rd of the camera frame. The travel time for the drone is set using a logarithmic mapping from the travel distance. As such, the drone can approach distant targets quickly, but still orbits smoothly when the user adjusts the viewing direction around the object. This speed tuning method is partially inspired by a similar virtual camera control technique [37].

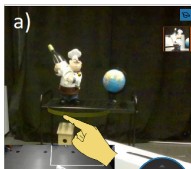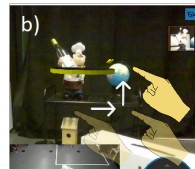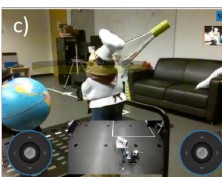

Figure 5: Interaction with the 360 viewpoint widget. (a) The user touches the area around the ring to activate the widget. (b) The user drags the finger to adjust the viewing angle and camera height. Upon releasing the drag, the drone navigates to the specified viewpoint.

### 4.5 Delayed Through-the-Lens Control

Through-the-lens camera control is a classic and intuitive camera manipulation technique in interactive graphics. It allows a user to move the camera by dragging one or more points in the current image to their desired new positions. For virtual environments, this procedure is usually performed in real-time. In the physical world, with incomplete scene information, performing this interaction is computationally heavy and error prone, and must deal with latency in the drone's movements [36]. By leveraging the estimated 3D position of the object-of-interest, in relation to the drone camera, we can calculate the required drone movement to achieve the specified viewpoint. Because drones cannot move in real-time like virtual cameras can, we delay the drone movements until the user has completed their gesture.

To use the technique, the user first rests two fingers on the drone camera view to freeze the current frame (Fig. 6a). The user then performs a two-finger pinch-and-pan gesture to transform the current frame to the desired viewpoint (Fig. 6b). The system then calculates a new drone position that can produce the desired viewpoint which the drone navigates towards.

### 4.6 Object-Centric Joysticks

Panning, zooming, and orbiting are standard camera control techniques in 3D modelling software. Prior research in interactive graphics proposed object-centric improvements, such as making the rotation center sticky to 3D objects [11]. We extend this method to drone camera control with *object-centric joysticks*. We remap the axis of traditional drone control joysticks to object-centric commands and add constraints to prevent manipulation errors. More specifically,

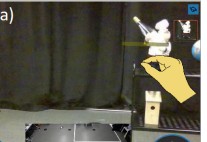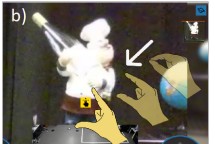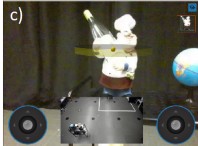

Figure 6: Adjusting the camera view using delayed through-the-lens control. (a) The user rests two fingers on the screen to freeze the current view. (b) A pan and zoom gesture on the frozen frame specifies the desired view.

the user can still push the left joystick up or down or push the right joystick left or right to pan the drone camera up, down, left, or right, respectively (Fig. 7a). However, under the object-centric constraints, the drone keeps the object of interest in its field-of-view during the pan movements. The up/down movement of the right joystick is mapped to an object-centric zoom, in which the drone aims its camera at the object of interest and moves closer or further away from it (Fig. 7b). When the left joystick is pushed to the left or right, the drone orbits around the object while aiming at its center (Fig. 7c).

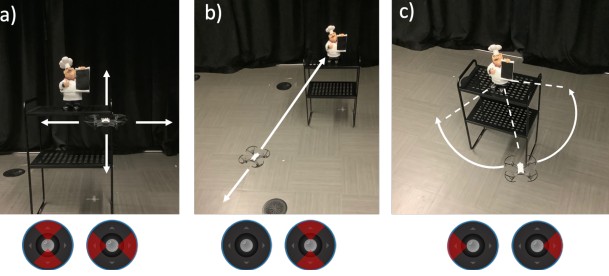

Figure 7: The object-centric joystick controls. Red areas indicate the joystick axes used. (a) Pan. (b) Zoom. (c) Orbit.

### 4.7 Manual Joysticks

In addition to the three object-centric navigation mechanisms, StarHopper also supports fully manual controls. This could be useful in cases where the user wishes to make slight adjustments to a viewpoint that the auto-pilot system navigated to. The manual control uses the same joysticks as the *object-centric joysticks*. The only difference is that the navigations are performed in world space, and not relative to the object-of-interest. Manual controls are enabled by toggling an icon on the interface (Figure 3e), or when no object-of-interest is selected.

### 4.8 Managing Objects of Interest

The object-of-interest list on the right of the interface (Fig. 3d) records the thumbnails of all previously registered objects of interest. The static images are taken from the drone camera during the object registration procedure. The user can tap on the thumbnail to set it as the object-of-interest, and the drone will turn towards it. A double-tap on the thumbnail will trigger the drone to approach that object using the auto-pilot system, until it reaches the default viewing distance. All previous objects are also shown as red spheres on the drone camera view, and can be tapped to be selected as the object-of-interest.

### 4.9 Implementation Details

To triangulate object positions, we acquire the camera parameters of the overview camera and the drone camera through a one-time calibration process. For prototyping purposes, we used the Vicon motion capture system for tracking the drone. This could potentially

be replaced with alternative internal or external tracking technologies with much lower cost and sufficient precision (e.g. [51]).

We use Z-axis-aligned (vertical to the ground) bounding cylinders as the approximate volumes for objects of interest. This representation has been chosen because it can be described with a small number of parameters and contains sufficient geometric information for common camera movements during visual inspection.

The two-view triangulation method in StarHopper is inspired by earlier works in model-free 3D selection [35, 38]. The two regions that a user selects in the overview camera and drone camera views are processed by an image segmentation algorithm (GrabCut [47]), to extract the foreground objects. We then back-project the geometric centers of the two extracted objects to two 3D rays. Their intersection can be approximated with the center of the line segment that connects the two closest points of the two rays. We use this approximated intersection point X as the geometric center of the bounding cylinder. We then compute the radius r and height h of the bounding cylinder using the following process:

- Let $M$ and $f$ be the external calibration matrix and focal length of the drone camera, calculate the position of the cylinder geometric center $X_c(x_c, y_c, z_c)$ in the camera coordinates using $X_c = MX$

- Let $w_b$ and $h_b$ be the width and height of the 2D bounding box of the object in the drone camera view, compute an approximate fit of the parameters r and h through matching the projection of the cylinder with the object 2D bounding box:

$$r = \frac{w_b z_c}{2f} \quad (1)$$

- Let $\theta$ be the tilt angle of the drone camera. Plug $r$ into the equation $fAB = h_b$ to solve for $h$, where

$$A = \left[ -\frac{h}{2}\cos\theta - r\sin\theta + y_c \quad \frac{h}{2}\cos\theta + r\sin\theta + y_c \right] \quad (2)$$

and

$$B = \begin{bmatrix} \frac{1}{-\frac{h}{2}\sin\theta + r\sin\theta + z_c} \\ \frac{1}{-\frac{h}{2}\sin\theta + r\sin\theta - z_c} \end{bmatrix} \quad (3)$$

While more advanced triangulation techniques could be used, we found that this method worked sufficiently well for the purpose of visual object inspection. If anything, the technique tends to slightly overestimate the size of the cylinder which was not problematic for navigation.

One challenge of the technique is that the object of interest may not be in the drone camera view, when the second gesture is needed to define the second region, for triangulation. To overcome this challenge, the drone turns and moves towards the region after the first gesture in the overview camera view is made. To do so, a ray is projected from the overview camera to the center of the initial region. The ray is then divided into a "coverage segment", where the object center could lie, based on a predetermined range of possible object heights. The drone then moves to the closest position, for which the drone camera's field-of-view can view the entire coverage segment. When navigating towards an object, we incorporated a simple mechanism adapted from the vector field histogram algorithm [4] to avoid obstacles.

We used a Ryze Tello drone in our prototype, in its default "slow mode" with a maximum speed of 6.7 miles per hour. Similar to other small consumer drones, it uses WiFi for communication, which introduces a small but noticeable delay in piloting and video streaming. The drone was controlled through a Python client based on the Ryze Tello SDK v1.3. The 4 degree-of-freedom velocity inputs were generated by 4 separate proportion-integral-differential (PID)

| StarHopper Navigation Mechanisms | Automation Level | Efficiency | Flexibility |
|---|---|---|---|
| 360 Viewpoint Widget | Automated | High | Low |
| Delayed Through-the-Lens | Semi-Automated | Medium | Medium |
| Object-Centric Joysticks | Semi-Automated | Medium | Medium |
| Manual Joysticks | Manual | Low | High |

Table 1: Properties of the four control mechanisms.

controllers. The user interface was implemented in C#/WPF and ran on a windows 10 laptop with a 4-core 2.7 GHz CPU, 16G RAM, and a touch screen.

## 4.10 Summary: Navigation Mechanism Properties

StarHopper consists of a set of four navigation mechanisms, ranging from fully automated to fully manual. Taken together, this suite of techniques allows users to perform both flexible and efficient scene inspections by leveraging their contrasting capabilities (Table 1).

To discuss this in greater detail, we use the Levels of Automation of Decision and Action Selection model [41] to define the automation level of a mechanism: A mechanism is considered as *automated* if it makes decisions for a user, and, as *semi-automated* if it requires a user to choose among available options, prior to executing the chosen option automatically.

The *360 viewpoint widget* is a highly *efficient* method as it enables a user to approach an object from any angle with one touch gesture. It has low *flexibility*, as the camera is constrained within a fixed orbit and at a fixed distance to the target center.

The *delayed through-the-lens control* is less efficient than the *360 viewpoint widget* as it can require more than one gesture to complete, and is designed for covering smaller distances and movements. For example, a target could be too far to be scaled with a single pinch gesture. It also supports a medium level of *flexibility* as the user can freely control the camera's 3D position and angle, but only through indirect 2D gestures.

Although *object-centric joysticks* enable convenient camera movements for visual inspection such as orbiting, the user needs to continuously and accurately push the joysticks. As a result, we consider it to have a medium level of *efficiency*. It also provides a medium level of *flexibility*, as the user can control the drone movements, but with the constraint of keeping the object in the field of view.

*Manual joysticks* are low in *efficiency* as the user needs to combine two or more commands to perform useful movements, like orbiting using the joysticks. We note that expert users familiar with the joystick mapping may find this technique efficient, but we consider it to be low efficiency, given the difficulty novice users would have with the technique. However, this technique does provide high flexibility, as it allows users to move the drone freely without any constraints.

From the analysis, we recognize the trend that a higher automation level increases efficiency but reduces flexibility. Taken together, the system offers the user both efficient and flexible navigation mechanisms (Table 1). The *360 viewpoint widget*, despite its high efficiency, lacks in flexibility and can be complemented by *delayed through-the-lens control*, *object-centric joysticks* and manual controls. A possible workflow for inspecting an object, is to quickly approach the object with the more efficient *360 viewpoint widget*, make fine adjustments with *delayed through-the-lens* or *object-centric joysticks* if needed, and freely explore the environment, perhaps to look for other potential objects of interest, using *manual joysticks*.

## 5 USER STUDY: NAVIGATION MECHANISMS

The above analysis shows the promise of combining both automatic and manual navigation mechanisms within a single interactive system. To evaluate the navigation mechanisms of StarHopper, we

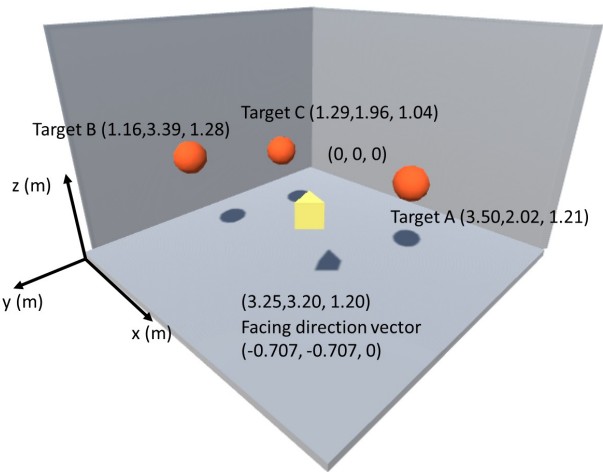

Figure 8: The environment consisted of 3 object locations (orange spheres) and an initial drone position (yellow prism).

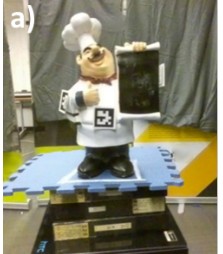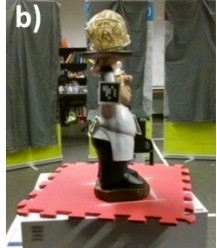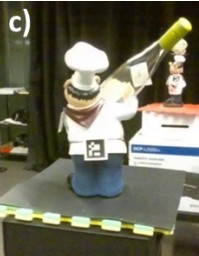

Figure 9: Reference photos of the target objects. (a) Front of ItemA. (b) Left of ItemB. (c) Back of ItemC. ArUco markers were attached to each side of the objects.

conducted a user study consisting of a remote object inspection task. We compared StarHopper to a baseline, consisting of conventional manual joystick controls. In tasks such as warehouse inventory management, most items are likely to be visited repeatedly, with occasional item addition or removal. For StarHopper, each object of interest can be reused for following revisits after a one-time registration. Therefore, in this study we left out the registration phase and focused on the actual navigation performance.

## 5.1 Participants

12 volunteers (7 female, $M_{age} = 26.3$, $SD_{age} = 4.4$) were recruited from the local community. Each received 20 dollars for their time. Two participants were familiar with quadcopter drone piloting but did not consider themselves experts. The others had no prior quadcopter drone piloting experience. Five of the twelve participants were frequent first-person-view (FPV) video game players, three had played FPV games but not frequently. The others had little FPV game experience.

## 5.2 Task and Stimuli

The task simulated a remote warehouse inspection scenario [52], where the participant, playing the role of a remote inspector, operated a camera drone to examine a number of key items in the environment. The test environment was set up in our lab, in which 3 target objects (*ItemA*, *ItemB*, *ItemC*) were positioned in the space (Fig. 8). In each trial, the participant was instructed to operate the drone to inspect one of the four sides (*Left*, *Right*, *Front*, *Back*) of an item using one of the two control interfaces, *StarHopper* or manual joysticks (*Manual*). The drone always started from the same initial position and orientation (Fig. 8). The front side of each target object faced the initial drone position. The participant was local, but a wall separator was used to prevent the participant from seeing the actual study environment, to simulate remote operation. As the study only aimed to evaluate the navigation performance of StarHopper, and not the two-view triangulation method, the target objects were predefined in the system.

For each control interface, the experimenter first demonstrated the operation of the interface. The participant then practiced piloting freely for 5 minutes, and performed 4 practice trials. To start a trial, the participant first tapped on a 'start' button. The item and side to inspect were indicated through a reference photo, shown to the participant before the trial began (Fig. 9). A three-second timer was used to start each trial. In a trial, the participant operated the drone

using the two virtual joysticks (*Manual*) or using any combination of the three object-centric navigation techniques for *StarHopper*. Thus, for the *StarHopper* condition, the user needed to tap on the target object prior to performing any navigation operations.

In order to complete the task, the participants needed to capture the specified side of the item, with a viewpoint similar to the reference photo. To quantify this criteria, we attached a ArUco fiducial marker [14, 45] to each side of the item (Fig. 9). When facing the marker, the drone camera could capture it and a program calculated the distance, d, between the drone camera and the marker center, and the angle, $\theta$, between the camera look direction and the marker surface normal. The drone was considered to be in the correct inspection position if $d < 800mm$ and $\theta < 45°$. The trial was completed if the drone remained in the correct position for 1.5 seconds. The 800mm distance was set to be smaller than the default viewing distance of the *360 viewpoint widget*, and thus, required the participants to make fine adjustments to the drone in all trials. If the drone collided with items in the environment, we would restart the trial. This only occurred three times across all participants.

On the study interface, a visual feedback icon was displayed to inform participants of the task state. The icon was grey if the target was out of sight, red if the target was too far or the viewing angle was too oblique, and green if the target was properly captured. They were also shown a progress bar advancing when the drone was in the correct inspection position. The participant was instructed to complete each trial as quickly as possible. When a trial finished, the drone automatically returned to its initial position. We recorded the completion time of each trial and the drone flight path. In addition, the participants were asked to fill in a NASA TLX worksheet after using each interface. A post-study questionnaire captured subjective preferences.

## 5.3 Design

A repeated measures within-subject design was used. The independent variables were *Interface* (*StrarHopper*, *Manual*), *Item* (*ItemA*, *ItemB*, *ItemC*), and item *Side* (*Left*, *Right*, *Front*, *Back*). The dependent variable was *Time*, defined as the time from when the trial started until the viewpoint was successfully captured, for the 1.5s duration. The presentation order of *Interface* was counter-balanced across participants. For each *Interface*, participants completed one trial for each of the 12 combinations of *Item* and *Side*, in a randomized order. Thus, the study consisted of 24 trials per participant, lasting a total duration of approximately 60 minutes.

## 5.4 Apparatus

The experiment was conducted in a 4.5 by 4.5 meters test environment in our lab. The participants were seated next to the test environment while performing the tasks, but their line of sight to the environment was blocked with barriers, to simulate remote operation. The target items were three chef figurines, each with distinctive

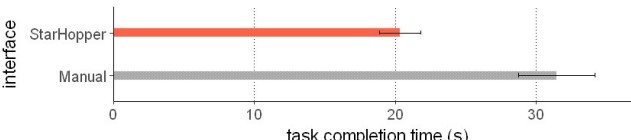

Figure 10: Mean task completion time of manual control and StarHopper. Error bars represent 95% CI.

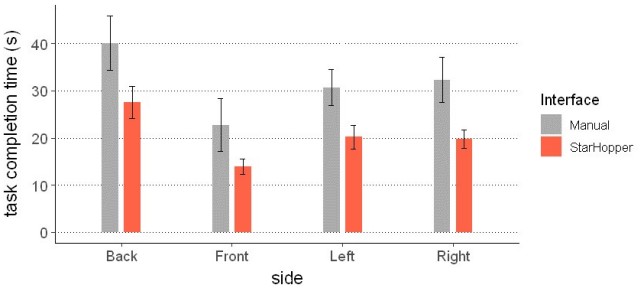

Figure 11: Mean task completion time for each Side. Error bars represent 95% CI.

features. Each of the four sides of the figurines had a 40mm AruCo marker.

For *StarHopper*, we used the same hardware and software configuration, as described in the Implementation Details section. For *Manual*, the drone was operated with the two virtual joysticks in the StarHopper interface, with the standard axis mapping of typical drone interfaces.

### 5.5 Results

A repeated measures analysis of variance with Greenhouse-Geisser correction was applied to analyze the performance data. We transformed the data with a logarithmic transform to satisfy the normality assumption before applying ANOVA tests. The transformation is only for statistical analysis purposes and we still report the descriptive statistics of the original data in the following results.

#### 5.5.1 Trial Completion Time

A main effect of *Interface* on *Time* was found ($F_{1,11} = 23.8, p < 0.001$), showing that it was significantly faster to complete the task with StarHopper than with egocentric manual control. Overall StarHopper was 35.4% faster than manual control (*StarHopper*: 20.33s, *Manual*: 31.45s), demonstrating a substantial gain in efficiency (Fig. 10).

We also found a significant main effect of *Side* on *Time* ($F_{2.8,30.4} = 31.9, p < 0.001$). Post-hoc pairwise comparisons using t-test with Bonferroni adjustments showed that inspecting the front side was significantly faster than the other three sides, and that inspecting the left or right side was significantly faster than the back side. This result showed that the participants' performance decreased as the navigation route complexity increased.

No significant interaction between *Interface* and *Side* was found. StarHopper demonstrated a consistent efficiency advantage over manual control (31% - 39%, Fig. 11), across the four sides. Even in the easier *Front* condition, participants still spent longer time with manual control, as they tended to make multiple short movements with pauses in between. A significant main effect for *Item* on *Time* was found ($F_{1.7,18.6} = 31.9, p < 0.05$). However, post-hoc tests did not find a significant difference between any of the two items. There was no interaction effect between *Item* and *Interface*.

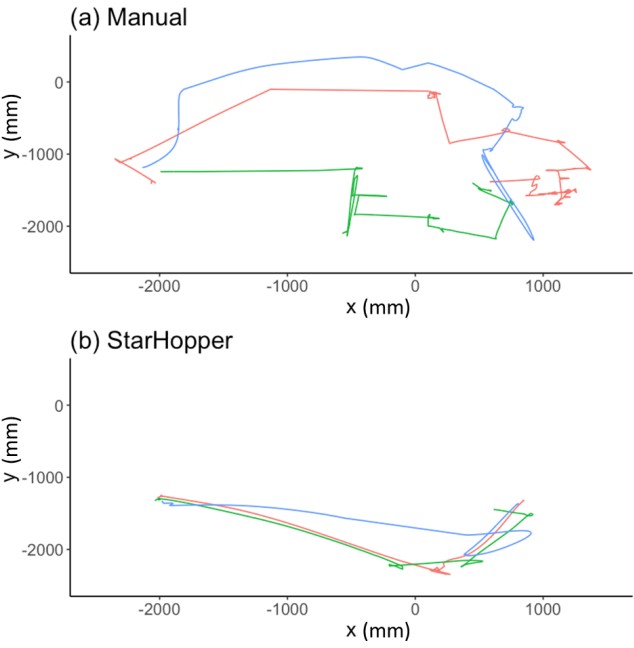

Figure 12: Three example flight traces for StarHopper and manual control, for the back side of ItemA. (a) Manual control. (b) StarHopper.

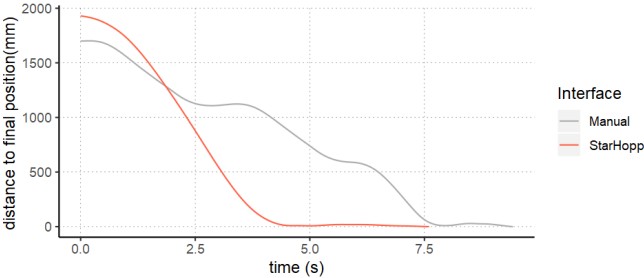

Figure 13: Distances to the final position as time progresses for the front side of ItemC for P11. With manual control, the user made short movements with intermittent pauses.

#### 5.5.2 Flight Traces

We further analyzed the flight traces to better understand the cause of the performance differences discussed above. For the manual condition, the traces tended to be connected line segments, suggesting that participants preferred keeping the orientation of the drone constant while moving. In contrast, for StarHopper, motion occurred in a more continuous nature, and was more likely to stay along an optimal path (Fig. 12).

One slightly surprising result from our analysis of trial completion time, was that *StarHopper* was faster even for the front views of the target objects. Our observations indicated that even for this easier manual task, participants would often start and stop when using the manual controls, to account for latency and to prevent any potential object collisions. Fig. 13 illustrates this effect, showing the distance to a final target position as a trial progresses, for both *Manual* and *StarHopper*.

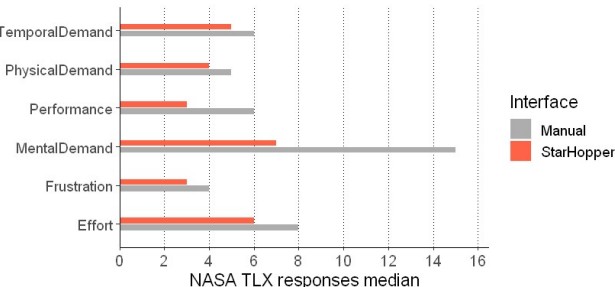

Figure 14: NASA-TLX responses along six dimensions. StarHopper was ranked significantly better for mental demand, physical demand, performance, and effort (lower scores indicate lower demand/difficulty).

### 5.5.3 Subjective User Preferences and Workload Perceptions

The post-study questionnaire responses showed that all participants preferred to use StarHopper over egocentric manual control for remote inspection. They commented that StarHopper was more intuitive and reduced navigation difficulty in comparison to the manual controls. P12 mentioned "*with joysticks people focus on navigation, with the other people focus on the object*". Nine of the twelve participants thought that virtual joysticks could add value to pure gestural interaction for drone navigation.

Paired Wilcoxon signed-rank tests on NASA TLX responses showed that StarHopper was ranked better than manual control for mental demand ($V = 56, p < .05$), physical demand ($V = 27, p < .05$), performance ($V = 36, p < .05$), and effort ($V = 61, p < .05$) (Fig. 14). Most notably, the median perceived mental demand for manual condition was more than twice as high as StarHopper.

## 5.6 Summary

Overall, the study results are very encouraging. The study confirmed the efficiency gain provided by the object-centric navigation techniques in StarHopper. Data further indicated that object-centric navigation could enhance remote inspection efficiency, for both simple and complex navigation routes. Users perceived lower workload using StarHopper, while subjective measures suggested an overall preference for object-centric controls.

## 6 DISCUSSION

In this section, we discuss the implications of the results from our study, clarify important considerations and limitations related to our work, and propose future lines of research.

## 6.1 Navigation Patterns

When demonstrating the three object-centric navigation techniques to the participants, we intentionally avoided displaying a fixed workflow. Participants were told they could use any workflow or combination of navigation techniques that they felt comfortable with. At the end of the warmup trials, many participants converged on a workflow in which they started with the *360 viewpoint widget* to approach the object, applied *delayed through-the-lens control* to adjust, and possibly *object-centric pan/orbit* ( *object-centric zoom* was rarely used) to fine-tune the final viewpoint. Overall, it was encouraging to see participants use a blend of the object-centric navigation techniques, leveraging their complementary attributes. Building Better Human-Automation Collaboration

StarHopper aims to explore the design of automation-human collaboration, in which automated systems execute tasks efficiently and humans adjust its behaviors, through semi-automatic or manual

methods [48]. However, a recurring theme in the participants' feedback was that they were hesitant to give the drone commands when it was in the midst of an automated movement, even though we had informed them is was ok to do so. Specifically, users commented that they did not want to interrupt the drone. Such observations are less likely to occur in a virtual navigation task, making it unique to physical viewpoint navigation. These breakdowns in automation-human communication can impede efficiency and highlights the importance of clearly communicating the intent and state of an automated system. This problem has drawn strong community interest (e.g. [33, 46]) and many proposed solutions can enlighten the next iteration of StarHopper. For local drone operation, augmented reality [56] could overlay the future flight path of the drone. Alternatively, icons [34] or subtle color changes [7], could be used display the intent or state. We believe that this is an interesting problem for future work.

## 6.2 Beyond Fixed Overview Cameras

While the area that the drone can explore with StarHopper is bounded by the field-of-view of the fixed overview camera, it can be greatly expanded if we replace the overview camera with a second, spatially-coupled camera drone [53]. Furthermore, StarHopper can also function outdoors with an accurate enough positioning system (e.g. RTK GPS) tracking both drones. In future work, we plan to explore new design opportunities enabled by these technical alternatives.

## 6.3 Limitations

We recognize that our study results are limited due to a number of technical and non-technical factors. For one, the control and video streaming latency displayed in our system is typical for small consumer drones, especially during remote piloting. The impact of latency on manual controls is likely to be greater than its impact on StarHopper. Additionally, the primitive obstacle avoidance ability of StarHopper restricted the maximum speed of the auto-piloting. We anticipate that incorporating more advanced collision avoidance algorithms could further increase its potential. The drone used in the study also only had one fixed forward-looking camera. With additional degrees-of-freedom, users would have more flexibility, but there would also be an extra burden due to increased control complexity.

As a first step in studying the performance of drone object-centric navigation, our study design is limited in several ways. The study was conducted in an artificial lab setting, with only three target items, all of similar dimensions, and placed in a regular rectangular formation. Such an environment cannot represent the complexity that a remote inspector may face in real-world environments. Additionally, the participants were also not truly remote as they could hear the noise from the drone. This could bring in extra information relevant to navigation.

Our study was focused on the efficiency of object-centric navigation. We did not formally study flexibility, or the performance and efficiency of the object-of-interest registration algorithm. We plan to further explore these aspects with future studies set in more complex environments. Finally, due to the difficulty of precise tracking in outdoor environments, our design and evaluation only touched on indoor inspection. As technology advances, we expect to extend our research to a more diverse range of settings, such as search and rescue in outdoor environments.

## 7 CONCLUSION

To remove the barriers of using drones as free-flying remote inspection platforms, we explored touch-based object-centric navigation for camera drones through our prototype system, StarHopper. An in-lab study showed that users were able to achieve notable efficiency improvement using StarHopper for remote visual inspection, in comparison to a baseline condition using a touchscreen joystick for manual control. A strength of StarHopper comes from its combined

use of a suite of automated, semi-automated, and manual control mechanisms to achieve efficiency and flexibility. In future work, we plan to empirically study users' mental models when working with automated camera drones, to understand how to build better human-automation collaboration for remote inspection. We are also interested in extending the usage of StarHopper to larger spaces and outdoor environments with a second drone as the overview camera.

Our object-centric navigation techniques took inspiration from classical techniques in interactive graphics. As 3D sensing, reconstruction, object recognition, and other related fields advance, more powerful techniques initially developed for the virtual world may be applicable to telepresence navigation, taking us even closer to the vision of unconstrained 'beyond being there' telepresence.

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
