# OpenReview forum: "StarHopper: A Touch Interface for Remote Object-Centric Drone Navigation"
_graphicsinterface.org/Graphics_Interface/2020/Conference — GI 2020_

### Official Review · AnonReviewer2 · 2020-01-07
**This paper introduces StarHopper, an application that uses computer vision techniques with touch input to support drone piloting with an object-centric approach. The system is interesting and validated via a user study that demonstrates its efficiency for finding and inspecting objects indoor.**

**Confidence:** 4
**Rating:** 9

**Review:**

The paper is clear and well written. Authors did a good job at motivating their work and the need for alternatives to current control methods for uavs used for video inspection tasks. However, I believe the introduction can be shortened since several parts do not bring much. For instance, the beginning of the introduction's second paragraph is not very relevant within the context of this paper. Only the last sentence adds to the argument I think.

The related work section covers and discusses well prior research in the field.

I really appreciated the design guidelines section even if it sounds more like design requirement to me than design guidelines per se. I have some concerns regarding the simple touch guideline argument. Authors state that it should be simple with already known gestures (I assume pinch, drag and swipe) and for more complex tasks (advanced path planning) an automated system should be able to do it. This should either be removed or clearly be defined since I do not believe that a "magic" algorithm will be able to infer users intentions. I would argue instead for basic and advanced interactions, possibly using another model that the object centric model for other type of tasks related to path planning. Finally, the last guideline (respect physical constraints) seems a bit in contradiction with minimizing the reliance on environmental data. I would suggest author to discuss more on how to combine these two guidelines.

The star hopper description is clear and easy to follow. The interactions and technical details are well presented. I have some concerns on how users can edit or delete an object of interest if they did not select the correct position or radius on the user interface. The size of the screen and the possibly large view of the scene seems very error prone to me. I would suggest authors to add some details on this issue.

The experiment is well presented and seems adequate to compare the StarHopper navigation techniques to a standard one. The results confirm that the systems helps user navigating efficiently in scenes to find and inspect objects.

Regarding the discussion, authors did a god job at highlighting limitations of their work. In particular, I also believe that making the state of the system visible and helping users to understand that they are able to act while the drone is operating is a very interesting direction.

Overall, I believe this is a very good paper that blends computer vision techniques, path planning and interaction. As such it clearly deserves an audience at GI. I would also suggest authors to submit a proposal to the CHI2020 workshop on human drone interaction to present this work or other related material. https://socialdrones.github.io/ihdi2020/

minor comments:
- no keywords provided
- Many drone systems are already use in industry such as monitoring paintings on wind turbines or aircrafts. Existing solutions are both manual and fully automated. It would be valuable for authors to include some examples in the paper.
- figure 13 is not understandable and grey print. authors can use colors with different luminance to counterbalance the problem.

---

### Official Review · AnonReviewer3 · 2020-01-08
**well written paper with good results, novelty and technical contribution is limited**

**Confidence:** 3
**Rating:** 7

**Review:**

##################
BY EXTERNAL REVIEWER
##################

This paper presents StarHopper, a system for semi-automatic drone navigation in the context of remote inspection. By using an external camera in addition to the drone camera, a set of interaction techniques for navigation are presented. Those techniques balance automatic navigation and manual input. In a user study, the authors show that their system is significantly faster than a manual 2-joystick control, and preferred by participants.

The paper is very well written, and the related work comprehensively covered and clear. While I liked the paper, there are a set of challenges that decrease my excitement, mostly in terms of conceptual and technical novelty, outlined below. In general, I am learning slightly positively towards the paper, mostly since the authors quantitatively show the superiority of advanced semi-automatic navigation techniques for drones, and that those can be combined in a single easy-to-use system.

On the positive side, the system and interaction techniques are very well described and clear. The paper provides a nice rational (design guidelines) for the system, and analyses it in terms of its levels of automation. The study is sound, and very well described. The results show that StarHopper, as a combination of multiple interaction techniques, is 35% faster than a conventional manual control, and preferred by users. The analysis of flight traces is interesting as well. Lastly, the paper provides an interesting discussion on the balance between automation and manual control, and how users would not interfere while the drone would perform a maneuver, even though they were explicitly instructed that this would be okay.

My main concern is regarding the novelty of the system. For me, the paper has two main contributions: 1) the idea of using an additional external camera for interaction, and 2) a set of interaction techniques. While the first contribution is novel but not used extensively, the interaction techniques are very similar to what is implemented in DJI's ActiveTrack system. The 360 Viewing Widget (circle around target) is included in DJI ActiveTrack (Trace mode + Joystick input). Object-centric joystick navigation is the same as DJI ActiveTrack (Spotlight mode). I acknowledge that 360 Viewing Widget is quasi-autonomous after the circle is set, this however I do not see as a major difference. This leaves the Delayed interaction technique, which is nice. The authors mention ActiveTrack in the related work section, and state that it assumes a single subject within the FoV and that it is less flexible in terms of navigation. I am not convinced by this argument. Circling with ActiveTrack is arguably more flexible (because less autonomy), and Spotlight and Object-centric joystick navigation are similar. The authors should make more explicit how the interaction techniques are different; and potentially condense the description of the techniques are less novel.

I think the paper has the right amount of technical description given its part in the contribution. On thing that should be clarified is if the authors only calibration the camera intrinsics and extrinsics, or also the 3D position of the external camera. Overall, I think the triangulation is a clever idea and can be replaced by some advanced techniques in the future.

In terms of presentation, while the paper is well written, it is quite verbose on some parts. The introduction focusses a lot on telepresence, but this is not addressed in the paper. I think the focus on remote inspection is fine and does not require such a lengthy intro. Similarly, parts of the interaction techniques could be condensed (manual navigation, managing the object of interest list), as well as the automation analysis. The focus on a touch interface is fine as well, but could be toned down. While touch is a dominant input modality, the interface would work equally well with a classical WIMP interface. Therefore, it is actually more general than 'only touch'.

In terms of the study results, it would have been interesting to see a break-down of how often the different modes were used. While the authors noted that a combination was used, I would have liked to read a more detailed analysis. If this data is available, I think it would be a valuable addition to understand the difference and usage between the individual modes better.

Overall, while this is a nice paper, I am somewhat on the fence due to the limited novelty, especially since large parts of the interaction techniques are already implemented in a commercially available and widely-spread system. Interesting directions would be to highlight the difference between outdoor navigation in nearly unconstrained space and very constrained outdoor space; and exploiting the external camera even more, eg for multiple or moving targets.

---

### Official Review · AnonReviewer1 · 2020-01-10
**Overall good paper**

**Confidence:** 4
**Rating:** 8

**Review:**

Summary: In this paper the authors outline a new drone control interface StarHopper that they have developed, that is combines automated and manual piloting into a new hybrid navigation interface. The automated part of the interface builds upon existing object-centric techniques, but gets rid of the assumption that the target object is already in the drone’s FOV by using an additional overhead camera. The interface itself consists of four modes – 360 degree viewpoint widget, delayed through-the-lens control, object-centric joysticks, and full manual joystick controls.

This hybrid interface was compared to a fully manual drone interface in a user study, and flight times for navigation tasks were compared between the two methods. Additionally, subjective user parameters such as effort, frustration, mental demand, etc. were compared. It was found that StarHopper outperformed the manual controls for this given task by a significant margin.

Review: I think the most exciting contribution from this paper is the way that each of the four modes included with StarHopper has a different strength, and they work very well used in series when completing a specific task, as is evidenced in the “Navigation Patterns” section. Shows that StarHopper is well-suited for the task for which the user study was run.  It is also encouraging that many of the users converged to a similar work flow after using the controls for such a short period of time – shows that they work together intuitively. It seems as though using well-known gestures such as pinch-and-zoom for the fine adjustments worked well and was intuitive for the users, based on the ratings in the subjective user preferences.

That being said, part of the motivation/design guidelines for StarHopper was to create an interface that performs well in a changing dynamic environment such as a warehouse. I think that future studies performed in a more realistic environment, including obstacles moving around in real-time would be a more useful assessment of the interface. I also wonder how well StarHopper would work in a typical warehouse environment that has tall aisles and other things that would obstruct the drones view – to register an object of interest it is necessary for it to not only be in the view of the overhead camera, but also within the drone’s FOV minus a rotation. This means that fully manual control would be necessary to approach a potential object of interest. And then if it moves in the dynamic environment, fully manual location may be necessary again. This could significantly reduce the speed improvements that are seen in the user study. The paper claims that previous object-centric solutions assume a subject that is already in the camera field-of-view, but it seems as if this is also true here minus a rotation.

Another concern is that StarHopper was compared against fully manual piloting rather than something more analogous (one of the other automatic techniques that had been mentioned in the paper). It seems almost self-evident that it will be more efficient to have the drone automatically aligned to the correct side of an object (using 360 viewpoint) with only fine tuning required.

It makes sense that in general navigation to the front of an object took less time than navigation to the sides, and navigation to the sides took less time than navigation to the back (ie. “This result showed that the participants’ performance decreased as the navigation route complexity increased.”). But I found it surprising that the speed benefits of StarHopper were not more pronounced for more complex routes such as navigating to the back of an object (““StarHopper demonstrated a consistent efficiency advantage over manual control (31% ~ 39%, Figure 11), across the four sides.””). Since the purpose of the 360 degree viewing widget is to increase the efficiency of the object-centric navigation rather than just flying to the object in the first place, why weren’t such benefits seen?

Next, the paper isn’t totally clear about the drone collisions – it mentions that a few took place, but doesn’t specify whether they were mostly under manual or automatic control. If StarHopper’s automatic controls lead to decreased collisions, this would be a big benefit over manual.

“NASA-TLX responses along six dimensions. StarHopper was ranked significantly higher for mental demand, physical demand, performance, and effort.” – doesn’t seem to be what the chart shows for performance, although I may just be misinterpreting how it is presented. To me the graph reads as StarHopper’s performance being rated lower.

Overall, I think that the proposed suite of navigation techniques alone are enough to make this contribution useful. But in future user studies I would prefer the technology compared to more analogous drone interfaces than pure manual, and also be used in a more natural environment.

---

### Meta-Review · Area_Chair1 · 2020-01-10

**Recommendation:** Accept
**Confidence:** 4

**Metareview:**

All reviewers were convinced by this work, by the results of the study showing that this drone interface outperforms existing techniques. All reviewers recommend acceptance. Congratulations!

---

### Decision · Program_Chairs · 2020-01-11

Accept